# Enhancing Soybean Salt Tolerance with GSNO and Silicon: A Comprehensive Physiological, Biochemical, and Genetic Study

**DOI:** 10.3390/ijms26020609

**Published:** 2025-01-13

**Authors:** Meshari Winledy Msarie, Nusrat Jahan Methela, Mohammad Shafiqul Islam, Tran Hoang An, Ashim Kumar Das, Da-Sol Lee, Bong-Gyu Mun, Byung-Wook Yun

**Affiliations:** 1Department of Food Security and Agricultural Development, College of Agriculture and Life Sciences, Kyungpook National University, Daegu 41566, Republic of Korea; 2Department of Applied Biosciences, College of Agriculture and Life Sciences, Kyungpook National University, Daegu 41566, Republic of Korea; methela.ag@nstu.edu.bd (N.J.M.); shafik.ag@nstu.edu.bd (M.S.I.);; 3Department of Agriculture, Noakhali Science and Technology University, Noakhali 3814, Bangladesh; 4Department of Environmental and Biological Chemistry, Chungbuk National University, Cheongju 28644, Republic of Korea; munbg@cbnu.ac.kr

**Keywords:** salt tolerance, ion homeostasis, ROS, gene expression

## Abstract

Soil salinity is a major global challenge affecting agricultural productivity and food security. This study explores innovative strategies to improve salt tolerance in soybean (*Glycine max*), a crucial crop in the global food supply. This study investigates the synergistic effects of S-nitroso glutathione (GSNO) and silicon on enhancing salt tolerance in soybean (*Glycine max*). Two soybean cultivars, Seonpung (salt-tolerant) and Cheongja (salt-sensitive), were analyzed for various physiological, biochemical, and genetic traits under salt stress. The results showed that the combined GSNO and Si treatment significantly improved several key traits, including plant height, relative water content, root development, nodule numbers, chlorophyll content, and stomatal aperture, under both control and salt stress conditions. Additionally, this treatment optimized ion homeostasis by enhancing the Na/K ratio and Ca content, while reducing damage markers such as electrolyte leakage, malondialdehyde, and hydrogen peroxide. The stress-responsive compounds, including proline, ascorbate peroxidase, and water-soluble proteins, were elevated under stress conditions, indicating improved tolerance. Gene expression analysis revealed significant upregulation of genes such as *GmNHX1*, *GmSOS2*, and *GmAKT1*, associated with salt stress response, while *GmNIP2.1*, *GmNIP2.2*, and *GmLBR* were downregulated in both varieties. Notably, the salt-sensitive variety Cheongja exhibited higher electrolyte leakage and oxidative damage compared to the salt-tolerant Seonpung. These findings suggest that the combination of GSNO and silicon enhances salt tolerance in soybean by improving physiological resilience, ion homeostasis, and stress-responsive gene expression.

## 1. Introduction

Soil salinization is a major challenge in contemporary agriculture, exacerbated by climate change and rising sea levels [1]. According to the Food and Agricultural Organization (FAO), salt-affected soils cover 424 million hectares of top soil (0–30 cm) and 833 million hectares of subsurface soil (30–100 cm) globally (representing 73% of the identified land area) [2]. Salinity stress, a significant abiotic factor, reduces agricultural productivity worldwide [3]. Excess salt concentration disrupts osmotic balance, causing water deficit, ion toxicity, and oxidative stress in plants [4]. Salinity affects many regions worldwide, especially in arid and semiarid areas with water scarcity and high evaporation rates. Salinity impacts 20% of arable land globally [5]. This includes both primary salinization, from soils with naturally high salt levels, and secondary salinization caused by human activities such as improper irrigation practices and poor water management [6].

Soybean (*Glycine max* L.), a globally cultivated legume, is a protein source for both animals and humans [7]. Native to East Asia, it is essential for global food security [8]. Although categorized as an oil seed with 23% oil content (dry basis), its high protein content (38–44%) also drives its global production [9]. Enhancing soybean yield requires effective management of biotic and abiotic stresses, including heavy metals, temperature, soil salinity, and drought [10]. Salinity is one of the most severe stresses, limiting production in arid and semiarid agro-ecological zones [11]. As a salt-sensitive crop, soybean reduces growth, yield, and nodulation efficiency under high salinity [12]. Research to mitigate salinity stress in soybeans is crucial because of this crop’s economic importance, sensitivity to salinity, and expanding salinized lands, as well as the potential of modern scientific techniques to effectively address these challenges.

Nitric oxide (NO) is a signaling molecule involved in several physiological processes. Common NO donors include sodium nitroprusside (SNP), S-nitroso glutathione (GSNO), and S-nitroso cysteine (CysNO) [13]. In salt-stressed crops, such as wheat, rice, and soybean, SNP treatment increases endogenous NO [14], promoting selective potassium (K^+^) and sodium (Na^+^) transport to maintain K^+^/Na^+^ homeostasis [15]. GSNO, a reservoir for NO, regulates cell signaling, inflammation, and antioxidant defense [16]. GSNO mitigates salt toxicity in tomato [14]. Similarly, silicon (Si) enhances plant tolerance to various abiotic stresses, including salinity [17]. It improves plant water use efficiency, boosts antioxidant capacity, and increases stress-related enzymes [18]. Si also helps maintain ion balance within plant cells by reducing Na^+^ uptake [19], promoting the selective uptake of beneficial K^+^ ions, thereby limiting the entry of toxic ions into the plant (Kumari et al. [20]. Salt stress disrupts water uptake and retention in plants, causing water deficit. Sharma et al. [21] suggested that Si enhances osmotic adjustment in plant cells by increasing the accumulation of compatible solutes, such as proline, sugars, and amino acids, which help maintain water balance and osmotic potential. Although extensive research has explored the individual effects of NO and Si on plant responses to abiotic stresses, their combined impact on root nodulation in soybean under salt stress remains unclear [22]. NO acts as a crucial signaling molecule that regulates plant responses to various stress conditions, including salinity [23]. It modulates root architecture, regulates stomatal behavior, and enhances antioxidant defense mechanisms, thus conferring salt stress tolerance [24].

However, the interactive effects of NO and Si on root nodulation in salt-stressed soybeans remain underexplored [25]. Tripathi et al. [26] emphasize the need to investigate potential synergistic or antagonistic interactions between NO and Si in this context to understand the complex regulatory mechanisms. Such knowledge could provide valuable insights for developing innovative strategies to enhance root nodulation and nitrogen fixation under saline conditions [27]. To gain deeper insights into the molecular mechanisms, we analyzed the relative gene expression of key salt stress marker genes, including the Na^+^/H^+^ antiporter gene (*GmNHX1)*, the salt overlay sensitive pathway gene (*GmSOS2*), and the K^+^ channel gene (*GmAKT1*). Additionally, we examined genes related to silicon transport, namely the Si transporter genes (*GmNIP2.1 and GmNIP2.2*), as well as the ferric leghemoglobin reductase precursor gene (*GmLbR*). By examining the combined impact of NO and Si on salt-stressed soybean growth and development, this study addresses knowledge gaps and enhances our understanding of the complex signaling networks and molecular mechanisms that confer soybean resistance to salinity stress. Furthermore, the findings of this study could inform strategies to improve soybean productivity in salt-affected agricultural systems, promoting sustainable food production and addressing salinity stress-related challenges. This study primarily aims to evaluate the compatibility and potential antagonistic effects of GSNO and Si in plants under stress conditions, with the long-term goal of exploring advanced formulations that enhance plant resilience, potentially utilizing nanoparticle technologies for efficient nutrient delivery and stress mitigation.

## 2. Results

### 2.1. Combined GSNO and Si Treatment Promoted Plant Growth

The combined GSNO and Si treatment enhanced plant growth and alleviated salt stress-induced adverse effects (Figure 1A). Our findings revealed significant differences between plants treated with and without salt. The salt treatment alone decreased plant height by 22.2% and 18.9% in Seonpung and Cheongja, respectively, compared to the controls (Figure 1B). However, combined treatment under salt conditions increased plant height by 13.9% and 18.2% in Seonpung and Cheongja, respectively, compared to the salt-only plant treatment. Additionally, the combined treatment improved the shoot FW by 44.6% and 57.5% in Seonpung and Cheongja, respectively (Figure 1D). Under control conditions, both GSNO and silicon, either alone or combined, significantly increased shoot dry weight in Cheongja; however, in the tolerant cultivar Seonpung, the shoot dry weight significantly increased with the combined treatment (Figure 1E). Exposure to salt increased the shoot dry weight by 30.3% and 48.3% in Seonpung and Cheongja, respectively, compared to plants treated with only salt (Figure 1E). No significant difference in the shoot RWC was observed in Seonpung; however, significant differences in the RWC were observed in Cheongja under salt conditions (Figure 1C).

### 2.2. Combined GSNO and Si Treatment Enhanced Root Traits

The GSNO and silicon treatments had a minor effect on root traits under salt-induced stress (Figure 2). In Seonpung and Cheongja, the salt treatment alone decreased the root fresh weight by 25.1% and 19.4% compared to the control (Figure 2E). The combined GSNO and Si treatment increased the fresh root weight to 50% and 40.1% in Seonpung and Cheongja, respectively, under salt stress (Figure 2E). In the salt-sensitive cultivar Cheongja, the dry weight significantly increased under salt conditions with the GSNO and silicon treatment independently or together (Figure 2F). Notably, the root RWC also significantly increased after the combined GSNO and silicon treatment in Seonpung (21%) and Cheongja (41%) compared to the salt treatment alone (Figure 2G).

Salt stress decreased the nodule quantity and weight of nodules (Figure 3). The combined treatment of GSNO and silicon increased the root nodule number by 33.3% (Cheongja) and 44.6% 9 (Seonpung) compared to the control under salt conditions (Figure 3D). The combined treatment also increased the nodule weight by 47.2% in Seonpung and by 38% in Cheongja by under salt conditions (Figure 3C). In the case of root surface area, root volume, and number of root forks and links, no significant differences were observed in both varieties under salt conditions (Appendix A).

### 2.3. Combined GSNO and Si Treatment Improved Chlorophyll Content and Stomatal Aperture

GSNO, a NO donor, and silicon significantly enhanced the pigment content and stomatal aperture (Figure 4). The chlorophyll content significantly decreased in Cheongja compared to Seonpung after exposure to salt (Figure 4A,B). Notably, GSNO alone significantly increased the chlorophyll content in both cultivars compared to the controls (Figure 4A,B): by 43.3% in Seonpung and 284.2% in Cheongja. Additionally, the combined GSNO and Si treatment under salt stress increased the chlorophyll levels by 61.6% (Seonpung) and 150.2% (Cheongja) compared to the salt-only treatment (Figure 4A). Chlorophyll b increased by 42% in Seonpung and 51.6% in Cheongja after the GSNO and Si treatment compared to the salt-only treatment, while salt stress alone closed the stomatal aperture (Figure 4D). The GSNO + Si treatment reduced salt stress and significantly promoted stomatal opening in both cultivars (Figure 4C,D).

### 2.4. Combined GSNO and Si Treatment Optimized Ion Homeostasis

Salt stress disrupts ionic homeostasis in soybean. Na^+^ accumulation and the Na/K ratio in soybean leaves dramatically increased under salt stress, while K accumulation significantly decreased, compared to the control group (Figure 5A–C). The GSNO + Si treatment significantly reduced Na accumulation (17.3% (Seonpung) and 16.63% (Cheongja)) and Na/K ratio (30.9% (Seonpung) and 51% (Cheongja)) compared to the salt-only treatment (Figure 5A,C). Conversely, K accumulation increased by 19.3% (Seonpung) and 70% (Cheongja) after the combined treatment, compared to that of the controls on exposure to salt (Figure 5B). While no significant change in Ca accumulation was observed in Seonpung, its accumulation significantly decreased in Cheongja after exposure to salt, and the GSNO + Si treatment increased Ca accumulation (Figure 5D). Si accumulation significantly increased with Si supplementation alone or combined in both cultivars (Figure 5E).

### 2.5. Combined GSNO and Si Treatment Reduced MDA and H_2_O_2_ Levels and Enhanced Cell Integrity, Proline, Water-Soluble Protein, and Antioxidant Activity

Electrolyte leakage indicates membrane injury caused by oxidative stress. Figure 6A,B show elevated ion leakage upon salt exposure. However, the GSNO + Si treatment reduced the damage and leakage. Proline, an osmoprotectant, significantly increased in Seonpung in comparison to Cheongja (Figure 6C) under salt stress. The application of GSNO and Si significantly increased the proline content in Seonpung (28%) and Cheongja (43%) compared to the solely salt-treated plants. Malondialdehyde (MDA) exhibited significantly higher levels with salt exposure. In Seonpung, lipid peroxidation in terms of MDA was significantly lessened when Si was applied alone and in combination with GSNO. However, in Cheongja, the combined application repressed the MDA content by 44.5% compared to the solely salt-treated plants (Figure 6D). Water-soluble protein exhibited a different trend; i.e., solely applied GSNO or Si increased soluble protein significantly more than the combined treatment (Appendix A). The H_2_O_2_ content was also significantly diminished in Seonpung (27.4%) and Cheongja (44.7%) with the combined application of GSNO and Si compared to the solely salt-treated plants (Figure 6E). The antioxidant APX, which decomposes H_2_O_2_, significantly increased in Seonpung (43%) and Cheongja (149.6%) with the combined treatment under salt stress (Figure 6F).

### 2.6. Combined GSNO and Si Treatment Modulates Gene Expression

To assess the impact of salt stress on gene expression, we conducted an RT-qPCR analysis. In *Glycine max* (soybean), the *GmNHX1* encodes a vacuolar Na^+^/H^+^ antiporter, and *GmSOS2*, a salt-sensitive gene, plays a crucial role in salt tolerance. *GmNHX1* transports Na^+^ and K^+^ into vacuoles, affecting plant development and salt tolerance. Our results revealed significant differences in the expression of *GmNHX1*, *GmSOS2*, *GmAKT1*, *GmNIP2.1*, *GmNIP2.2*, and *GmLBR1* in Seonpung and Cheongja varieties (Appendix A). *GmNHX1* was markedly enhanced with the combined application of GSNO and silicon (7.24- and 5.34-fold, respectively) in Seonpung and Cheongja under salt treatment (Appendix A) compared to the control. The *GmSOS2* gene was also enhanced in Seonpung (2.2-fold) and Cheongja (1.6-fold) compared to the group of plants that were only salt-treated (Appendix A). *Lsi1* homolog, the silicon transporter gene Nodulin 26-like Intrinsic Protein (NIP), *GmNIP2.1,* and *GmNIP2.2* were downregulated by the treatments (Appendix A). However, *GmAKT1* expression was found to be enhanced significantly in the combined treatment (7.2- and 2.8-fold, respectively) in Seonpung and Cheongja under salt stress compared to the control (Appendix A). Moreover, relative gene expression was notably higher in Seonpung than Cheongja. However, *GmLBR* was downregulated, with no significant difference in response to the GSNO and Si treatment, whether alone or combined, in both cultivars under salt stress (Appendix A).

Figure 7 illustrates the gene expression profiles of key salt-responsive genes analyzed under various treatment conditions. In the Cheongja cultivar, the combined GSNO and Si treatment under salt stress resulted in the highest expression levels of *GmNHX1* and *GmSOS2*, peaking at a value of 9. This suggests a synergistic effect between GSNO and Si in enhancing salt tolerance (Figure 7A). Additionally, *GmAKT1* was moderately upregulated in response to this treatment. In Seonpung, the salt + GSNO + Si treatment resulted in the highest expression levels, particularly for *GmSOS2*, which reached an expression value of 12, followed by *GmNHX1* and *GmAKT1* (Figure 7B). These results indicate a strong salt tolerance response from the GSNO and Si combination.

## 3. Discussion

Salinity significantly limits crop yield and quality by causing an osmotic imbalance and ionic toxicity, impairing growth and photosynthesis. Si regulates natural plant hormone levels and essential protein expression. Our study indicates that salt stress significantly stunts plant growth (Figure 1). However, the combined GSNO and silicon treatment at the V1 stage alleviated this salt-induced negative effect in both cultivars. At the V1 stage, plants are more sensitive to treatments as secondary and tertiary root initiation begins. Additionally, they exhibit relatively uniform growth, allowing for consistent and reliable application of treatments across all samples. A comparatively lower reduction in plant height was recorded in tomato seedlings under 150 mM salt with the application of GSNO [14]. An increase in plant height may also be connected to a higher production of nitric oxide and reactive oxygen species (ROS), which are important signaling molecules [28]. Additionally, the RWC increased with the GSNO and silicon treatment, alone or combined, under salt stress (Figure 1C). Cultivating salt-tolerant crops such as Seonpung has positive environmental implications. Salt-affected soils are often unsuitable for traditional crops, leading to land degradation. Using these salt-tolerant varieties, we can reclaim and sustainably utilize these lands without causing further damage to the ecosystem [29]. In a recent study, it was reported that the application of exogenous silicon produces a notable increase in the levels of physiologically active endogenous gibberellin [30], which is responsible for increased plant growth.

Under salt stress, GSNO increases chlorophyll content in plants by mitigating salt-induced stress [31] through osmotic regulation, antioxidant protection, NO signaling, nutrient uptake, and stomatal control. This results in healthier and more chlorophyll-rich plants, which are better equipped to conduct photosynthesis even in saline environments. The combined application of GSNO and Si under salt conditions resulted in the highest increase in chlorophyll content (Figure 4A,B). This might be because NO is an essential signaling molecule, while Si is a beneficial element. Furthermore, when combining these two, the plant showed tolerance to salt due to the different function of application of this treatment. Exposure to high levels of salt can result in an uneven spatial arrangement of stomata in plants, as well as a reduction in stomatal density, directness, and conductance. Stomatal opening and closing are essential for maintaining physiological processes, for example transpiration, and help in osmotic balance. Stomatal closure is a common adaptation to salt stress [32]. Our result revealed that the combined GSNO and Si treatment increased stomatal aperture compared to the control treatment in both Seonpung and Cheongja cultivars (Figure 4C,D). Additionally, under the control conditions, the stomata was open, while under salt conditions, the stomata was closed (Figure 4D and Appendix A).

The development of root nodules is a distinctive characteristic of leguminous plants. The symbiotic nitrogen fixation within these root nodules serves as a primary nitrogen source for synthesizing crucial biomolecules [33]. Both the quantity and quality of these root nodules play a crucial role in determining the overall nutritional well-being of the entire plant. The decreased nodule numbers and weights of soybeans under salt stress stem from the harmful effects of salt ions on rhizosphere biota and soil pH, while salt stress reduces nodulation factors in legumes [34]. Additionally, it has been observed that salt stress can impede the transport of nutrients to the nodules. Our result demonstrated that salt stress affected the nodulation of soybean and decreased root nodule number and biomass (Figure 3A–D, Appendix A). This is because salt stress decreases the aerobic respiration of nitrogen-fixing bacteria, reduces the leghemoglobin content in the root nodules, and depletes the energy source required for nitrogen fixation [35,36]. Additionally, when plants were exposed to salt stress prior to the nodulation process, there was a significant reduction in the number of nodules formed. This pre-nodulation salt stress likely interfered with the early stages of nodule development, ultimately hampering the plant’s ability to form a sufficient number of nodules. The application of the combined GSNO and Si treatment significantly improved root morphological traits under salt stress conditions, as evidenced by increases in the total root length, root surface area, average root diameter, and number of root tips, root forks, and root links (Figure 2). Similar improvements in total root length, average diameter, and surface area under salt stress have been reported in other studies after the sole application of nitric oxide or silicon [37,38].

Salinity induces hyperosmotic stress and ion imbalance in plants, prompting them to store inorganic ions (Na^+^, Cl^−^, and K^+^) in their vacuoles to maintain osmolality, which prevents osmotic imbalance by synthesizing osmoprotectants such as proline, promoting continuous growth [39]. Our result showed that the exogenous GSNO and Si treatment suppressed ion accumulation (Na, K, Si, and Ca) in the plant tissues under salt stress conditions (Figure 5). This implies that these treatments might impact the plant’s response to salt stress and could potentially both have beneficial effects and reduce detrimental ones, depending on the specific context and goals of the research. The treatments, particularly the combination of GSNO and Si, seemed to maintain ionic balance in both tolerant (Seonpung) and susceptible (Cheongja) cultivars under salt stress (Figure 5). While salt treatment increased Na^+^ levels as expected, the combined treatment counteracted this effect, suggesting a protective role against salt-induced ionic imbalance (Figure 5A). Salt stress typically causes an elevated presence of sodium ions (Na^+^) in the cytoplasm of plants, along with an increased Na/K ratio [40]. This situation can lead to sodium toxicity. When the external environment has both high Na^+^ levels and elevated pH values due to salt stress, it creates a steeper concentration gradient for Na^+^ and hydrogen ions (H^+^) inside and outside the plant cells. This gradient encourages the excessive accumulation of Na^+^ within the cells, adversely impacting plant growth and development [41]. Maintaining adequate potassium (K^+^) levels is crucial for salinity tolerance, with salt-tolerant crops favoring K^+^ absorption over sodium (Na^+^) [42]. The intracellular penetration of Na^+^ can impair K^+^ function, leading to enzyme cofactor imbalances. Additionally, calcium (Ca^2+^) plays a vital role in strengthening cell walls, modulating membrane permeability, regulating stomatal function, and activating enzymes, thereby reducing oxidative stress and enhancing plant resilience in saline environments [43]. Ca^2+^ inhibits Na^+^ buildup through mechanisms such as the Salt Overly Sensitive (SOS) pathway and Mitogen-Activated Protein Kinase (MAPK) cascades, regulating osmotic pressure and protecting against salt stress [44]. Our investigation revealed that administering S-nitrosoglutathione (GSNO) and Si significantly reduced Na content under salt stress while increasing K and Ca levels (Figure 5A–D). This treatment also significantly minimized the Na/K ratio compared to salt exposure alone (Figure 5C). Sustaining low cytosolic Na and high K along with high Ca concentrations is essential for proper cellular activity, especially under salt-stressed conditions, ultimately influencing overall plant health [43].

Plants often experience oxidative stress, leading to changes in biochemical markers such as electrolyte leakage, soluble protein content, MDA, and H_2_O_2_. These changes reflect how plants cope with stress by activating antioxidative defense mechanisms and repairing oxidative damage. These markers help assess the impact of salt stress on plant health and the effectiveness of antioxidant responses in mitigating stress-induced damage. To counteract the detrimental effects of salt stress-induced osmotic stress, plants increase osmolyte levels in the cytosol and organelles [45]. This study observed a similar accumulation pattern of proline and increased water-soluble protein under salt conditions (Figure 6B,D). The response of total soluble proteins to saline stress has been well documented [46]. Additionally, proline plays a crucial role in protecting the photosynthetic machinery and storing energy during NaCl-induced stress [47]. Research indicates that proline accumulation enhances N_2_ fixation in *Medicago truncatula* plants and has the capacity to decompose ROS under salt stress conditions [48,49]. Furthermore, the application of nitric oxide (NO) to salt-stressed chickpea plants significantly increases the levels of total soluble proteins, proline, and soluble sugars, potentially offering enhanced protection to the plants under stress. The protective function of these osmolytes under NO treatment is consistent with findings from various studies [41,50,51].

Salt stress induces oxidative stress, increasing MDA and electrolyte leakage in both cultivars under salt stress, with higher levels in the sensitive cultivar Cheongja (Figure 6A,D). The combined application of GSNO and silicon significantly reduced the MDA and EL under salt stress in both varieties compared to their respective control. This reduced Na uptake, mitigated toxic symptoms in the cell physiology and cellular membrane by scavenging ROS, and helped in maintaining cell integrity. Similar findings were also recorded for lowering EL by solely applied NO or Si in rice and soybean [52,53] and combined NO + Si treatment in mustard [54] under different abiotic stress conditions. Reduced MDA was also recorded in salvia [55], salt-sensitive tomato [56], and legumes [51,57], with the exogenous application of NO or Si independently. It has been claimed that AsA-GSH cycle metabolism is responsible for the elimination of H_2_O_2_. APX reduces H_2_O_2_ into water using AsA as the electron donor; the resultant dehydroascorbate (DHA) is cycled back to AsA using reduced GSH as the electron donor; and the oxidized glutathione (GSSG) is converted back to GSH via NADPH-dependent GR [58]. In our study, elevated levels of hydrogen peroxide (H_2_O_2_) were observed under salt exposure. This increase in H_2_O_2_ was decomposed by ascorbate peroxidase (APX) activity, which was enhanced by the combined application of GSNO and Si together (Figure 6E,F). A lowering of H_2_O_2_ has been reported in salvia, tomato, and soybean under salt stress with solely applied NO or Si [41,52,53,55]. Furthermore, APX activity was exhibited to increase with NO and/or Si application in different plant species under salt exposure [50,53,56,59].

The regulation of ion concentration and homeostasis is crucial for plant growth under salt stress. Plants have developed various mechanisms to tolerate high salt concentrations, such as storing excess salt in older tissues or sequestering it in vacuoles. Sodium ions (Na^+^) are transported into vacuoles via transmembrane Na^+^/H^+^ antiporters, driven by the proton motive force generated by proton pumps in the tonoplast [60]. The *GmNHX1* gene, which encodes a Na^+^/H^+^ antiporter, may actively contribute to the plant’s response to salt-induced stress. *GmNHX1* is primarily located in vacuolar membranes, where it sequesters Na+ absorption in root vacuoles to prevent toxic accumulation, while limiting the transport of Na^+^ to leaves [61]. This compartmentalization of Na^+^ is essential for maintaining cellular ion homeostasis and ensuring plant survival under high-salinity conditions. Several investigations have found that salt stress activates the Salt Overly Sensitive (SOS) pathway. Salinity stress promotes the Ca^2+^ signaling pathway, stimulating the synthesis of threonine/serine kinase encoded by the *SOS2* gene [62]. Consequently, we investigated the relative expression of the *GmNHX1* and *GmSOS2* genes. Enhanced expression of both *GmNHX1* and *GmSOS2* genes was observed with the combined GSNO + Si treatment in both Seonpung and Cheongja varieties compared to their respective solely salt-treated plants (Appendix A). Higher salt tolerance or susceptibility was seen in soybean plants that overexpressed or were *GmNHX1* knockouts, respectively [63]. Additionally, compared to wild-type plants, *GmNHX1*-transformed plants displayed a higher Na^+^ efflux rate and maintained a higher K^+^/Na^+^ ratio [12,63]. The *SOS2* gene has been reported to be upregulated in mung bean under salt stress conditions, and this expression is further significantly enhanced upon the application of Si [64]. On the other hand, the K^+^ channel gene *GmAKT1* was observed to be upregulated by treatments in both cultivars; however, it was more enhanced in the tolerant cultivar Seonpung (Appendix A). Under salt stress, the *AKT1* gene has been reported to be enhanced in wheat by regulating K^+^ uptake [65]. The overexpression of *GmAKT1* might enhance potassium uptake, indirectly improving soybean salinity resistance [66]. This resistance is potentially achieved by upregulating *GmNHX1* expression, which facilitates sodium sequestration from the cytoplasm into the vacuole during salt stress.

Furthermore, we investigated the silicon transporter genes *GmNIP2.1* and *GmNIP2.2. GmNIP2.1* and *GmNIP2.2* were downregulated in both the control and salt environments (Appendix A). *MsNIP2.1* overexpression increased salt tolerance in *Medicago sativa* [67]. In pomegranate, *NIP2.1* was reported to contribute to its survival under saline conditions [68]. Si was taken up by the roots and distributed to various organs regulated by the *low silicon* (*Lsi1*, *Lsi2*, and *Lsi6*) and *NIP2* genes in mung bean and cucumber with Si application under salt conditions [64,69]. However, in dicot species like soybean, *Lsi1* homologs *GmNIP2.1* and *GmNIP2.2* exhibit lower expression and Si deficiency when supplemented with Si [26,70,71], which is in line with our result (Appendix A). Exogenous Si and GSNO may reduce the need for additional Si uptake as the available Si is likely sufficient for mitigating salt stress. The leghemoglobin gene, *GmLbr*, also showed downregulation upon receiving the treatments (Appendix A). Under saline conditions, nitrogenase activity drastically reduces and inhibits leghemoglobin in legumes [36]. The downregulation of *GmLBR* might be linked to a shift in metabolic or physiological priorities under salt stress conditions, where the plant focuses more on managing ionic balance and less on nitrogen fixation processes. As the heatmap illustrates, both varieties demonstrated increased gene expression under salt stress with GSNO and Si; however, *Seonpung* exhibited notably higher overall expression levels, particularly for *GmNHX1* and *GmSOS2*. This stronger response in *Seonpung* may indicate a genetic predisposition for enhanced salt tolerance, making it a more suitable candidate for high-salinity environments. The elevated expression levels of *GmNHX1* and *GmSOS2* in *Seonpung* under the combined treatment suggested that these applications can further bolster this cultivar’s inherent salt tolerance mechanisms. Additionally, *GmAKT1* underscores the broader role of GSNO and Si in regulating ion transport under stress conditions. The upregulation of *GmAKT1* implies that GSNO and Si might also enhance potassium homeostasis during salt stress, contributing to overall plant resilience. The application of GSNO and Si under salt stress conditions results in the downregulation of Si transporter genes and leghemoglobin, possibly indicating a shift in the plant’s resource allocation. However, the upregulation of *GmNHX1* and *GmSOS2* suggested that the combined treatment enhanced the plant’s ability to tolerate salt stress by improving ionic homeostasis.

## 4. Materials and Methods

### 4.1. Synthesis of GSNO and Chemicals Used

GSNO was prepared by reacting equimolar concentrations of sodium nitrite (NaNO_2_, Sigma-Aldrich, Darmstadt, Germany) with reduced glutathione (GSH, L-glutathione reduced, Sigma-Aldrich, Darmstadt, Germany) pre-dissolved in hydrochloric acid (1 N). For Si treatment, sodium metasilicate (Na_2_SiO_3_) was used.

### 4.2. Plant Material and Growth Condition

The seeds of two soybean cultivars, Seonpung (salt-tolerant) and Cheongja (salt-sensitive [72], were obtained from the Plant Functional and Genomics Laboratory, Kyungpook National University. The seeds were sown in 6 cm × 40 cm polypropylene pipes containing sandy soil; the experiment followed a 4 × 2 factorial design with three replications. The pipes were placed in a greenhouse at KNU with a controlled photoperiod (16 h light/8 h dark cycle) and temperature (28 °C ± 2 °C). Sprinkler irrigation was applied until the commencement of the treatments. Treatments commenced at the V1 stage (first fully expanded trifoliate) and lasted for 10 consecutive days, applying GSNO (0.1 mM), Si (2 mM), and NaCl (150 mM) via soil drenching and foliar application [73,74] with 100 mL per plant. The treatments included the following: control, salt, GSNO, GSNO + salt, Si, Si + salt, GSNO + Si, and GSNO + Si + salt for each cultivar. At 28 days of age, fresh leaf samples were collected from four replications, snap-frozen in liquid nitrogen, and stored at −80 °C for further analysis.

### 4.3. Plant Height and Relative Water Content

Plant height was measured using a centimeter scale and weighed using a BSA 3223S-CW Max 320 digital weighing scale (NBCHAO, Guangzhou, China). The relative water content (RWC) was estimated [75]. Briefly, leaf and root samples were gathered and gently cleaned three times with water, and the fresh weight (FW) was measured. Following hydration of the samples for two hours, turgor weight (TW) was calculated. The samples were then dried for at least 48 h at 65 °C to obtain a constant weight. The RWC was determined with the following formula:RWC (%) = ((FW − DW)/(TW − DW)) × 100

### 4.4. Analysis and Sampling of Roots

Plants were uprooted, the soil was carefully removed, and the roots were gently cleaned and then dried with paper towels. The roots were scanned using an EXP12000XL scanner (Epson, Nagano, Japan) and root morphology was analyzed using the analysis program WinRHIZO^TM^ Pro (Regent Instruments Inc., Quebec City, QC, Canada) [76].

### 4.5. Chlorophyll Content and Stomatal Aperture

Chlorophyll content was measured using a SPAD-502 Chlorophyll Meter (Konica Minolta, Osaka, Japan). For each leaf, three random positions were measured and the average value was calculated. Chlorophyll and carotenoid content were determined from the uppermost fully opened trifoliate leaf of each plant [77]. Briefly, 100 mg of fresh leaves was submerged in 20 mL of dimethyl sulfoxide (DMSO), incubated for 4 h at 65 °C, and then cooled to room temperature. Absorbance was recorded at 663, 645, and 470 nm using a UV spectrophotometer (UV-1280, Shimadzu, Kyoto, Japan). Chlorophyll a, b, and carotenoid contents were calculated using Arnon’s equations below:$${Chla}_{\left(mg/g\;fresh\;weight\right)}=\left[\left(12.7\times A_{663}\right)-\left(2.69\times A_{645}\right)\right]\times \frac{Vol}{W\times 1000}$$$${Chlb}_{\left(mg/g\;fresh\;weight\right)}=\left[\left(22.9\times A_{645}\right)-\left(4.68\times A_{663}\right)\right]\times \frac{Vol}{W\times 1000}$$$${Total\;Chlorophyll}_{\left(mg/g\;fresh\;weight\right)}=(20.08\times A_{645}+8.02\times A_{663})\times \frac{Vol}{W\times 1000}$$$${Car}_{\left(mg/g\;fresh\;weight\right)}=\frac{1000A_{470}-1.82Chla-85.02{Chl}_{b)}}{198}\times \frac{Vol}{W\times 1000}$$
where V = volume of extract (mL), W = fresh weight of the samples (g), and *Car* = Carotenoids.

To measure stomatal aperture, a thin coat of transparent nail polish was applied on the abaxial surface of the leaves and allowed to dry. Transparent scotch tape was softly applied over dry nail polish to create an impression of the stoma. After carefully removing the tape, it was placed on microscopic slides. The prepared slides were examined using a multimedia vedio microscope (MST-APT600, MASTER EDU, Gyeonggi, Korea) equipped with a 40× objective, images were captured using MEMS software 3.0, and ImageJ software (version java8) was used for image analysis.

### 4.6. Ion Contents

ICP-MS measurements (Optima 7900DV, PerkinElmer, Waltham, MA, USA) were used to estimate Na^+^, K^+^, Ca^++^, and Si content [52]. Freeze-dried samples (200 mg) were homogenized with nitric acid (5 mL; d 1.38, Fujifilm wako, Osaka, Japan) and H_2_O_2_ (3 mL; Duksan, Republic of Korea), and then the supernatant was diluted with 3% HNO_3_. The sample was filtered through a 0.45 m syringe filter (Sterlitech, Auburn, WA, USA) and injected into a coupled plasma mass spectrometry analyzer [52].

### 4.7. Water-Soluble Protein, Electrolyte Leakage, Proline, Malondialdehyde (MDA), H_2_O_2_, and APX Content

Total protein was quantified using Bradford protein assay reagent (Thermo Fisher Scientific, Waltham, MA, USA) by following the manufacturer’s instructions provided in the kit manual [78]. Electrolyte leakage (EL) was estimated following the protocol described by [79], with a slight modification [52]. Briefly, three leaves, each with two leaf disks measuring 1 cm in diameter, were collected from the same plant and counted as one replication. The disks were rinsed with distilled water and placed in a tube containing deionized water (10 mL) for 2 h at room temperature. A portable conductivity meter (HURIBA Twin Cond B-173, Kyoto, Japan) was used to measure the electrical conductivity (L1) of the solution. The samples were then autoclaved for 20 min at 120 °C and equilibrated at 25 °C to determine the final conductivity (L2). EL was calculated using the formula below:EL (%) = L1/L2 × 100 

Proline content was assessed using the method described by Batelli et al. in 2007. Plant tissue (0.5 g) was homogenized in 3% sulfosalicylic acid, and 2 mL of the supernatant was reacted with an equal volume of glacial acetic acid and ninhydrin, incubated at 100 °C for 1 h, and cooled in an ice bath to stop the reaction. To extract proline, 4 mL of toluene was added and vortexed, and the upper organic phase was carefully separated. Absorbance was measured at 520 nm, with toluene as the blank. Proline content was quantified and expressed as micromoles per gram of FW.

To measure MDA and H_2_O_2_, 250 mg leaf samples were homogenized with 1.5 mL trichloroacetic acid (1%) and centrifuged at 11,500 rpm for 15 min at 4 °C. For MDA, the supernatant was mixed with reaction buffer (1:2 ratio) containing 20% TCA and 0.5% thiobarbituric acid (TBA) and incubated at 95 °C for 1 h, immediately cooled in an ice bath for 5 min, and centrifuged at 10,000× *g* for 10 min [80]. Absorbance was measured at 532 nm and 600 nm using a spectrophotometer (Shimadzu UV-1280, Kyoto, Japan). The MDA concentration was determined using a 155 mM^−1^ cm^−1^ extinction coefficient and expressed as micromoles of MDA g^−1^ FW. For H_2_O_2_ measurement [81], to 3 mL of the reaction mixture, we added 100 mM potassium phosphate buffer (500 µL), 1M potassium iodide (2 mL; added last), and leaf extract supernatant (500 µL). The mixture was incubated for 1 h in the dark followed by 20 min at room temperature and the absorbance was measured by using a spectrophotometer at 390 nm. A standard curve was used to calculate H_2_O_2_ levels. APX activity was detected at 290 nm and an extinction coefficient of 2.8 mM^−1^ cm^−1^. The assay mixture consisted of 20 µL of enzyme extract, 0.1 µM EDTA, 50 mM phosphate buffer, 0.5 M ascorbate, and 1 mM H_2_O_2_ in a final volume of 1 mL [82].

### 4.8. RNA, cDNA Synthesis, and Relative Gene Expression Analysis

RNA was extracted from frozen leaf samples using TRIzol reagent (MRC, Cincinnati, OH, USA) following the manual’s instructions. cDNA was synthesized using the SolGent DiaStar™ RT-Kit (SolGent, Daejon, Republic of Korea). Quantitative real-time PCR (qRT-PCR) was carried out to evaluate relative gene expression. Synthesized cDNA was utilized as a template for conducting qRT-PCR using a BioRad CFX Duet Real-time PCR system (Singapore) and Solg^TM^ 2× Real-Time PCR Smart mix including SYBR^®^ Green I. The PCR process involved three steps, comprising 30 cycles. The first step was conducted at 95 °C for 15 min to activate the polymerase, followed by denaturation at 95 °C for 20 s. The second step involved annealing at 58–60 °C according to the melting temperature for 40 seconds. And the third step involved extension at 72 °C for 30 s. To confirm the specificity of the primer pair and assess the amplicon’s identity, a melting curve analysis was performed, ranging from 60 to 95 °C. Finally, the results for relative gene expression, obtained from the qRT-PCR analysis, were compared with phenotypic data from salt-stressed soybeans. We evaluated the relative expression of *GmNHX1*, *GmSOS2*, *GmAKT1*, *GmNIP2.1, GmNIP2.2,* and *GmLbR.* The transcripts of *GmNHX1*, *GmSOS2*, and *GmAKT1* were analyzed in leaf samples, while the transcripts of *GmNIP2.1*, *GmNIP2.2*, and *GmLbR* were examined in root samples. As an internal control, *GmActin7* was used [83]. The primer lists are listed in Appendix A.

### 4.9. Statistical Analysis

Data were analyzed using R-Studio (Boston, MA, USA) version 4.3.1 software for analysis of variance (ANOVA). A post hoc Duncan Multiple Range Test (DMRT) was used to compare group means at *p* ≤ 0.05. Data visualization of the results was performed using GraphPad Prism version 9.3.0 (San Diego, CA, USA).

## 5. Conclusions

The synergistic use of GSNO and silicon significantly enhances soybean salt tolerance by promoting plant growth, enhances root traits, improves chlorophyll content and stomatal aperture, and optimizes ion homeostasis. Additionally, it reduces electrolyte leakage, MDA, and H_2_O_2_ levels while enhancing proline, water-soluble protein, and antioxidant activity. The upregulation of stress-responsive genes further supports this approach. Integrating GSNO and Si offers a sustainable strategy to mitigate soil salinity, contributing to the knowledge of plant biology and establishing a foundation for future research on stress mitigation and crop performance enhancement in challenging environments. The authors recommend integrating silicon and GSNO into a synergistic complex like a nanoparticle to offer a promising strategy for sustainable agriculture. This approach harnesses silicon’s ability to enhance plant stress resistance and GSNO’s role in regulating growth and defense mechanisms, promoting healthier crops with reduced dependency on chemical inputs.

## Figures and Tables

**Figure 1 ijms-26-00609-f001:**
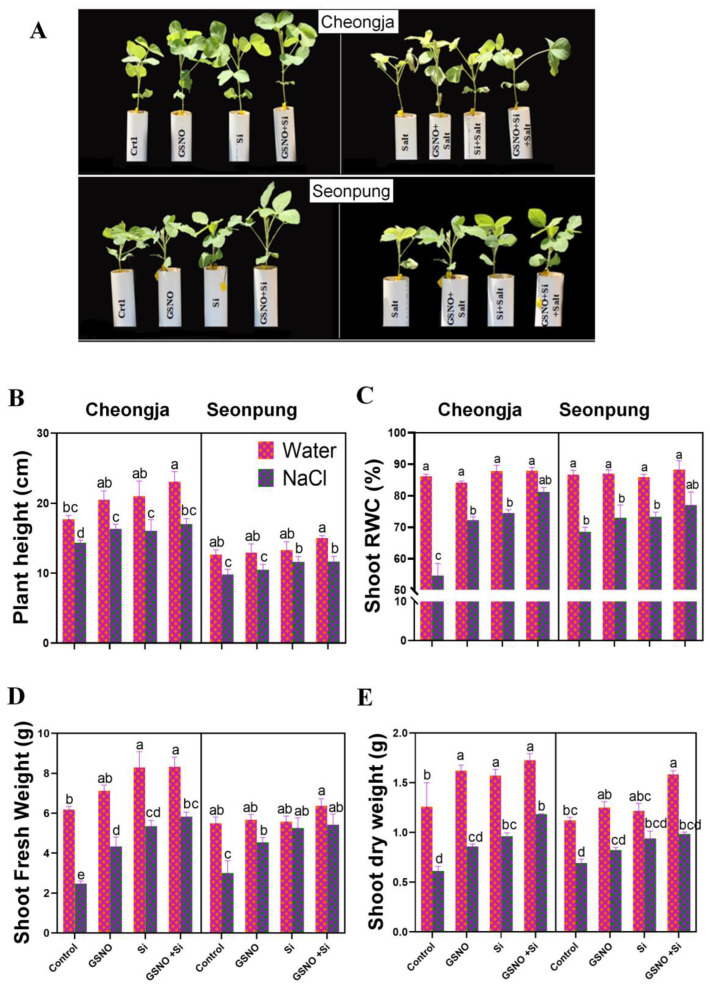
The combined application of GSNO and Si enhanced plant growth in the soybean cultivars Seonpung and Cheongja under salt stress. (**A**) The plant phenotype of Cheongja and Seonpung, (**B**) plant height, (**C**) shoot relative water content, (**D**) shoot fresh weight, and (**E**) shoot dry weight. Plants were treated with control, GSNO, Si, and GSNO + Si under two factors: water and salt conditions. Sampling was performed at 28 days of age. The bar graph displays the standard error of the mean, with each data point representing the average of three replicates. The letters on the bars indicate significant differences determined using the DMRT at *p* ≤ 0.05. The same letters indicate no significant differences.

**Figure 2 ijms-26-00609-f002:**
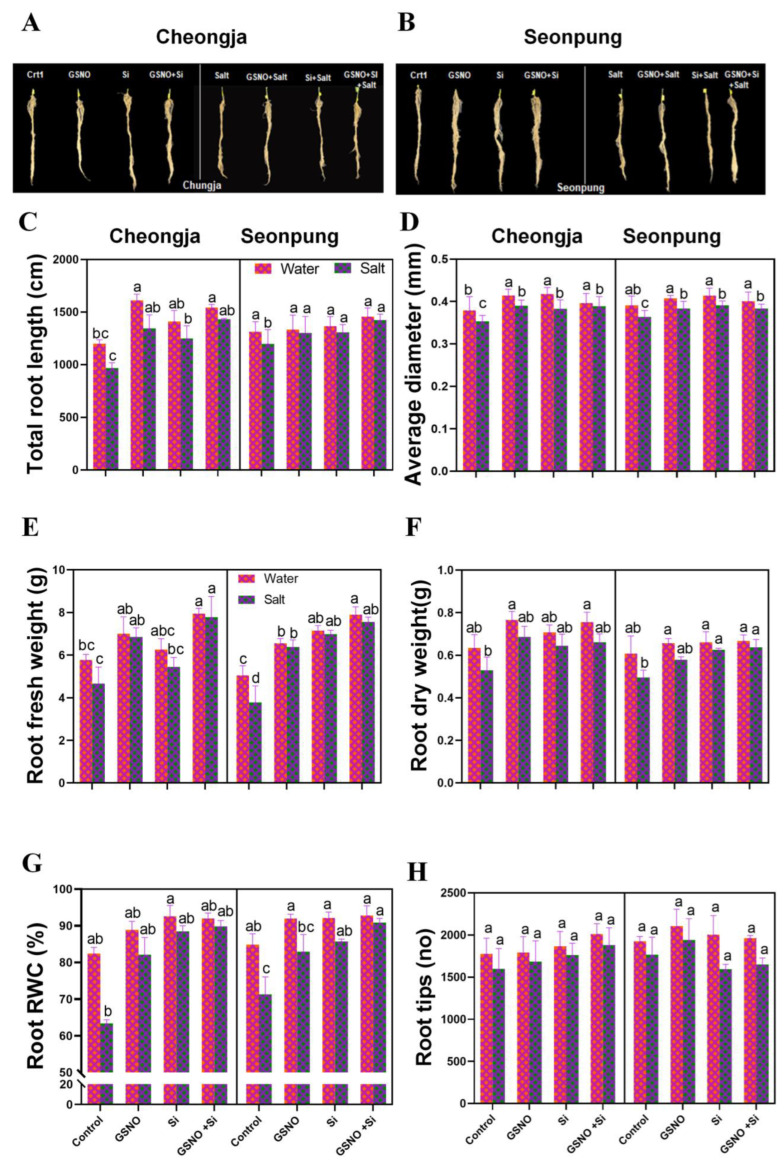
The combined GSNO and Si treatment improved root traits in the soybean cultivars Seonpung and Cheongja under salt stress. (**A**) The root morphology of Cheongja, (**B**) root morphology of Seonpung, (**C**) total root length, (**D**) average root diameter, (**E**) root fresh weight, (**F**) root dry weight, (**G**) root relative water content, and (**H**) number of root tips. The bar graph displays the standard error of the mean, with each data point representing the average of three replicates. The letters on the bars indicate significant differences determined using DMRT at *p* ≤ 0.05. The same letters indicate no significant differences.

**Figure 3 ijms-26-00609-f003:**
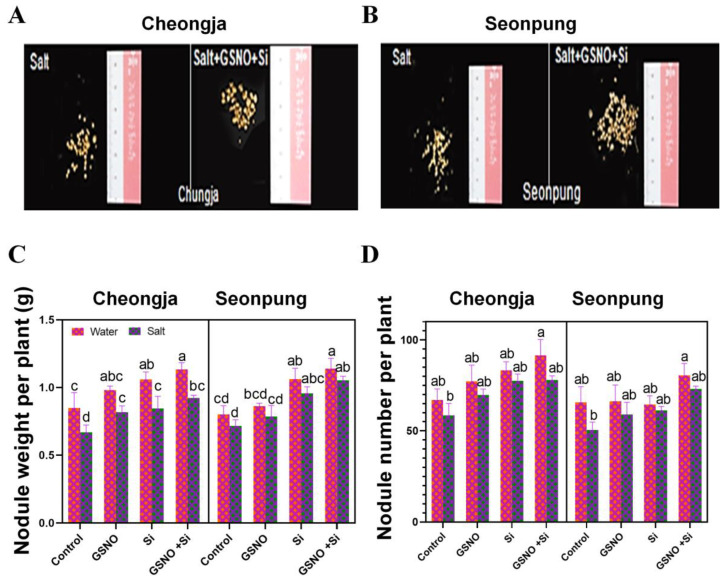
Combined GSNO and Si treatment increased nodule numbers in the soybean cultivars Seonpung and Cheongja under salt-induced stress. (**A**) Nodules in Cheongja, (**B**) nodules in Seonpung, (**C**) nodule weight, and (**D**) nodule number. A centimeter (cm) ruler in A and B is shown in for scale reference. The bar graph displays the standard error of the mean, with each data point representing the average of three replicates. The letters on the bars indicate significant differences determined using DMRT at *p* ≤ 0.05. The same letters indicate no significant differences.

**Figure 4 ijms-26-00609-f004:**
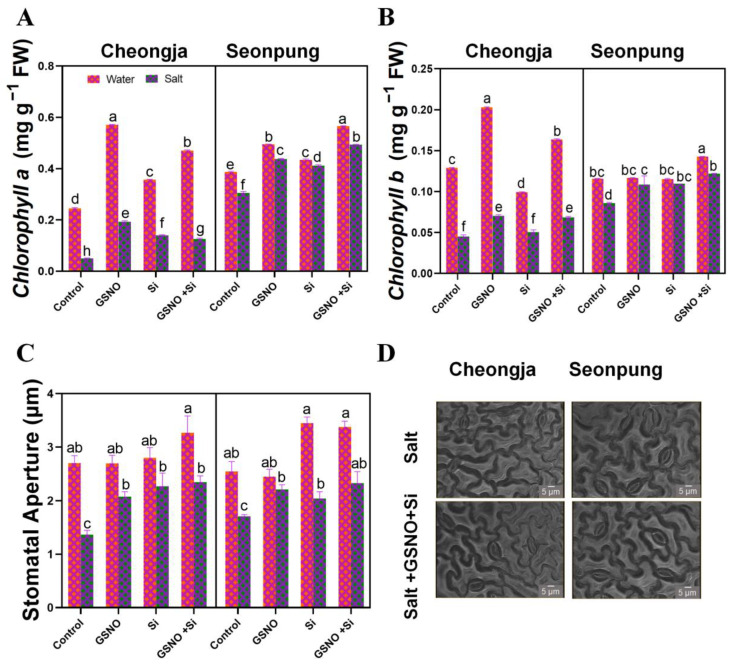
Combined GSNO and Si treatment promoted plant growth in Seonpung and Cheongja under salt stress. (**A**) *Chlorophyll a*, (**B**) *chlorophyll b*, (**C**) stomatal aperture, and (**D**) leaf stomata under a light microscope. The scale bar represents 5 µm. The bar graph displays the standard error of the mean, with each data point representing the average of three replicates. The letters on the bars indicate significant differences determined using DMRT at *p* ≤ 0.05. The same letters indicate no significant differences.

**Figure 5 ijms-26-00609-f005:**
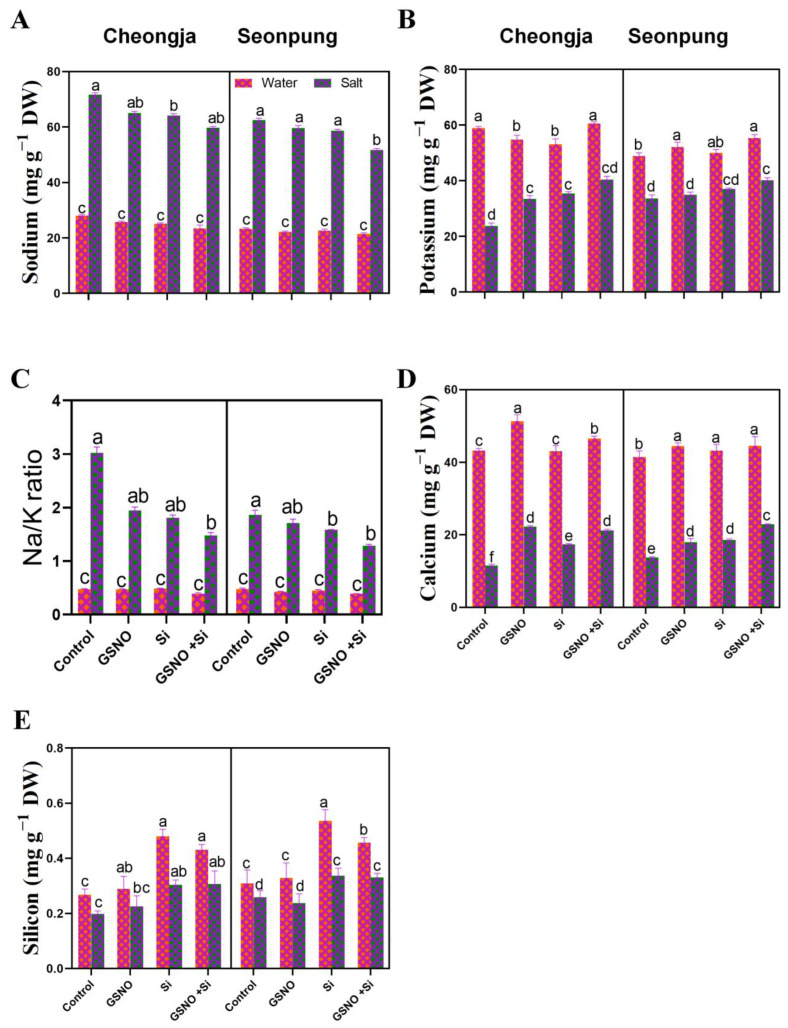
The combined GSNO and Si treatment regulates ion homeostasis in Seonpung and Cheongja under salt stress. (**A**) Na content, (**B**) K content, (**C**) Na/K ratio, (**D**) Ca content, and (**E**) Si accumulation. The bar graph displays the standard error of the mean, with each data point representing the average of three replicates. The letters on the bars indicate significant differences determined by DMRT at *p* ≤ 0.05. The same letters indicate no significant differences.

**Figure 6 ijms-26-00609-f006:**
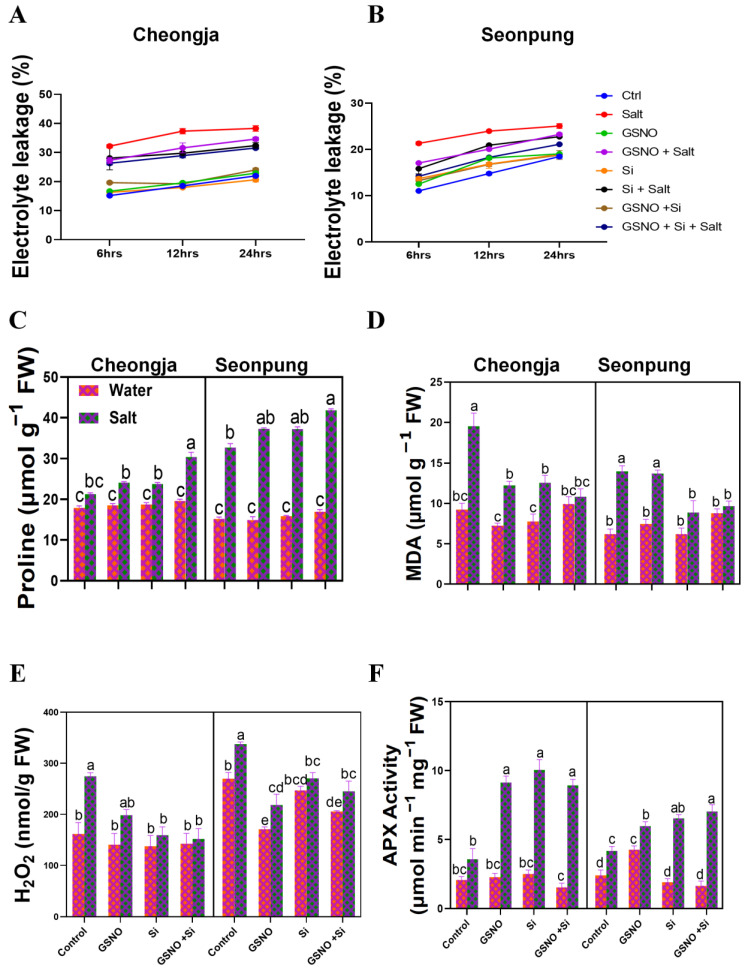
The combined GSNO and Si treatment reduced cell leakage, MDA, and H_2_O_2_ levels by increasing APX activity and proline accumulation in Seonpung and Cheongja under salt stress. (**A**) Electrolyte leakage in Cheongja, (**B**) electrolyte leakage in Seonpung, (**C**) proline content, (**D**) MDA content, (**E**) H_2_O_2_ content, and (**F**) APX activity. The bar graph displays the standard error of the mean, with each data point representing the average of three replicates. The letters on the bars indicate significant differences determined using DMRT at *p* ≤ 0.05. The same letters indicate no significant differences.

**Figure 7 ijms-26-00609-f007:**
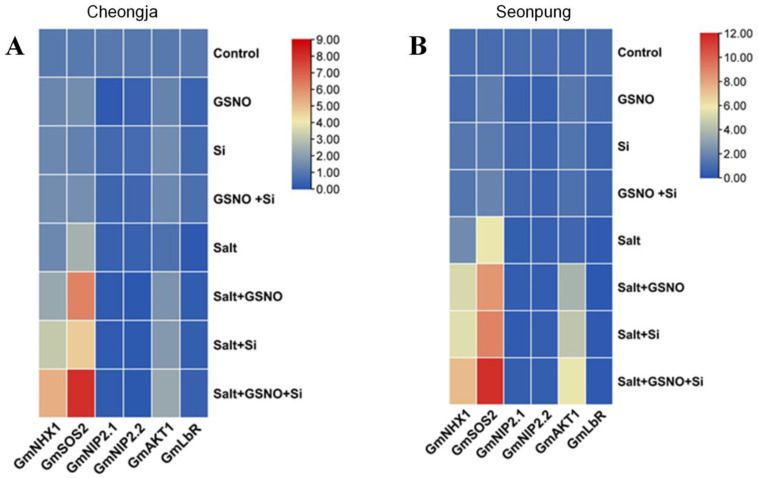
Heatmap of relative gene expression in response to treatments: (**A**) Cheongja and (**B**) Seonpung. The heatmap was generated using TBtools based on the mean expression values from three biological replicates. Red indicates high expression levels, while blue represents low expression levels, reflecting the correlation strength between gene expression and treatments. The genes analyzed include *GmNHX1*, *GmSOS2*, *GmNIP2.1*, *GmNIP2.2*, *GmAKT1*, and *GmLbR*, with treatments such as GSNO, Si, salt, and their combinations.

## Data Availability

Data is contained within the article and Appendix A.

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
