# Peer review of "Enhancing Soybean Salt Tolerance with GSNO and Silicon: A Comprehensive Physiological, Biochemical, and Genetic Study"

_ijms, 2025, doi:10.3390/ijms26020609_

Round 1

Reviewer 1 Report

Comments and Suggestions for Authors

(Line 133-134) Don’t use Korean font also check in the whole manuscript.

Use cultivar or variety in the whole manuscript.

Gene expression analysis revealed significant upregulation of stress-responsive genes GmNHX1, GmSOS2, and GmAKT1, while GmNIP2.1, GmNIP2.2, and GmLBR were downregulated in both varieties. Why do authors only select these stress-responsive genes please mention this in the introduction.

(Line 70) Soybean, being a salt-sensitive crop, experiences a decline in growth, yield, and nodulation efficiency under high salinity conditions. Please add a reference.

Figure 7 Please add a variety names and treatments in all figures (A-F)

Add only Figure 8 in the main manuscript and move figure 7 in the supplementary file.

(Line 139-141) Treatments commenced when plants were at V1 stage (first fully expanded trifoliate) by both soil drenching and foliar application for 10 consecutive days. GSNO (0.1 mM), Si (2 mM) and NaCl (150 mM) (Pruthi et al., 2024; Tripathi et al., 2022)

Why authors only select V1 stage for treatment please mention in the introduction and discussion section with reference for clear understanding.

Figure -1 please mention the treatment and sampling time of the plant in the figure legend.

Comments on the Quality of English Language

English is not my mother tongue but I strongly suggest correcting the grammar by a native specialized speaker.

Author Response

We sincerely appreciate your insightful comments and suggestions, which have significantly contributed to improving the quality of our manuscript. Each of your comments has been thoroughly addressed, and the corresponding revisions have been incorporated into the manuscript as detailed in our response letter.

We are deeply grateful for the time and effort you have dedicated to reviewing our work and providing constructive feedback. Your expertise has been invaluable in enhancing the clarity and scientific rigor of our study.

Thank you once again for your thorough review and kind consideration.

Sincerely,

Reviewer 2 Report

Comments and Suggestions for Authors

Article title: Enhancing Soybean Salt Tolerance through GSNO and Silicon: A Comprehensive Physiological, Biochemical, and Genetic  Study

In general, the experiment is very good and very well written. Some observations to improve the manuscript.

ABSTTRACT

Please, verify if the amount of words is according to the journal’s requirement.

INTRODUCTION

1. L 75 – 105: Paragraph is too big. Exclude lines 102-105. It’s repetitive!

MATERIAL AND METHODS

2. Need improvements. Insert the folloing informations:

- How was done the irrigation? Applied volume, irrigation frequency, etc.

- How long was the experiment? How many days plant was collected for analysis?

- What was the experimental design used?

- Was it a factorial scheme 4 x 2? Inform it!

RESULTS AND DISCUSSION

Results and discussion sections are ok!

CONCLUSION

The conclusion section is okay! However insert any recommendation for future works.

Author Response

(The authors gave the same response as above.)

Round 2

Reviewer 1 Report

Comments and Suggestions for Authors

As per my previous comment, the authors have made all corrections. Now, this can be accepted for publication.

Comments on the Quality of English Language

The authors have made all the necessary corrections.

Reviewer 2 Report

Comments and Suggestions for Authors

After reading the resubmitted material, I recommend the manuscript for publication in its current form.

Best regards.